# *ABCB1* Amplicon Contains Cyclic AMP Response Element-Driven *TRIP6* Gene in Taxane-Resistant MCF-7 Breast Cancer Sublines

**DOI:** 10.3390/genes14020296

**Published:** 2023-01-23

**Authors:** Petr Daniel, Kamila Balušíková, Radka Václavíková, Karolína Šeborová, Šárka Ransdorfová, Marie Valeriánová, Longfei Wei, Michael Jelínek, Tereza Tlapáková, Thomas Fleischer, Vessela N. Kristensen, Pavel Souček, Iwao Ojima, Jan Kovář

**Affiliations:** 1Department of Biochemistry, Cell and Molecular Biology, Third Faculty of Medicine, Charles University, 100 00 Prague, Czech Republic; 2Toxicogenomics Unit, National Institute of Public Health, 100 00 Prague, Czech Republic; 3Laboratory of Pharmacogenomics, Biomedical Center, Faculty of Medicine, Charles University, 323 00 Pilsen, Czech Republic; 4Department of Cytogenetics, Institute of Hematology and Blood Transfusion, 128 00 Prague, Czech Republic; 5Department of Chemistry, Institute of Chemical Biology & Drug Discovery, Stony Brook University—State University of New York, Stony Brook, NY 11794, USA; 6Department of Cell Biology, Faculty of Science, Charles University, 128 00 Prague, Czech Republic; 7Department of Cancer Genetics, Institute for Cancer Research, Oslo University Hospital, 0310 Oslo, Norway; 8Department of Medical Genetics, Institute of Clinical Medicine, Faculty of Medicine, University of Oslo, 0424 Oslo, Norway

**Keywords:** *TRIP6*, cAMP response element, gene amplification, *ABCB1*, breast cancer, CpG methylation, MCF-7

## Abstract

A limited number of studies are devoted to regulating *TRIP6* expression in cancer. Hence, we aimed to unveil the regulation of *TRIP6* expression in MCF-7 breast cancer cells (with high *TRIP6* expression) and taxane-resistant MCF-7 sublines (manifesting even higher *TRIP6* expression). We found that *TRIP6* transcription is regulated primarily by the cyclic AMP response element (CRE) in hypomethylated proximal promoters in both taxane-sensitive and taxane-resistant MCF-7 cells. Furthermore, in taxane-resistant MCF-7 sublines, *TRIP6* co-amplification with the neighboring *ABCB1* gene, as witnessed by fluorescence in situ hybridization (FISH), led to *TRIP6* overexpression. Ultimately, we found high *TRIP6* mRNA levels in progesterone receptor-positive breast cancer and samples resected from premenopausal women.

## 1. Introduction

Thyroid hormone receptor interactor 6 (*TRIP6*, 7q22.1) gene encodes for highly conserved 50 kDa protein (476 amino acid residues). TRIP6 is structurally organized into an N-terminal proline-rich domain, Crm-1-dependent nuclear export signal (NES), and three tandemly arrayed LIM domains (lin-11, Isl-1, and mec-3) followed by PDZ-binding motif TTDC (PSD95, Dlg1, ZO-1) at its C terminus [1,2]. TRIP6 belongs to small zyxin family (zyxin, ajuba, lipoma-preferred partner 1, LIM domain-containing protein 1, Wilms tumor 1-interacting protein, filamin-binding LIM protein 1) due to the presence of LIM domains [3].

Due to various modules, TRIP6 interacts with a plethora of partners extensively summarized elsewhere [4]. TRIP6 resides in the cell cytoplasm, where it accumulates at focal adhesions [5] and adherent junctions [6,7,8], the sites associated with the actin cytoskeleton. Additionally, TRIP6 has been described to shuttle between the cell nucleus and cytoplasm [2]. The N-terminally truncated isoform of TRIP6 (nTRIP6) entirely localizes in the cell nucleus, where it promotes transcription of myogenesis regulatory genes [9,10,11]. Recently, Shukla et al. demonstrated that half of the *TRIP6* knock-out mice developed hydrocephalus as a result of impaired ciliogenesis in ependymal brain cells [12]. In conclusion, TRIP6 participates in multiple cellular processes, cell motility [6,13,14,15], signaling [15,16], regulation of transcription [5,9], and telomere protection [17]. Thus, TRIP6 represents an attractive molecule in cancer research.

Several lines of evidence underpin high *TRIP6* expression in MCF-7 cells. The Human Protein Atlas displays *TRIP6* mRNA as the most expressed in MCF-7 cells “https://www.proteinatlas.org/ENSG00000087077-TRIP6/cell+line (accessed on 19 January 2023)” [18]. Analysis of publicly available single cell transcriptomic data “https://bcatlas.tigem.it/tigem/dibernardo/AIRC_atlas_32_ccls/?ds=Atlas_32_ccls&gene=TRIP6 (accessed on 19 January 2023)” showed ubiquitous *TRIP6* expression in 32 evaluated breast cancer cell lines [19]. Nevertheless, BT549, MCF-7 and HCC70 cells expressed more *TRIP6* mRNA than any other breast cancer cells in the study. In line with this finding, Zhao et al. reported high *TRIP6* expression in several breast cancer cell lines [20]. Furthermore, we revealed *TRIP6* overexpression in paclitaxel-resistant MCF-7/PacR subline [21], yet the molecular mechanism(s) that drive *TRIP6* expression in MCF-7 cells as well in paclitaxel-resistant cells have not been described in detail.

To date, a few studies have investigated the regulation of *TRIP6* expression by microRNA (miRNA), in particular, miR-138-5p [22], miR-485-3p [23] in neural stem cells, and miR-7 in colorectal cancer [24]. Recently, Gou et al. demonstrated that tocopherol α transfer protein-like TTPAL protects TRIP6 from ubiquitin-mediated degradation in colorectal cancer [25].

Breast cancer is the most diagnosed cancer worldwide [25]. It exhibits a heterogeneous nature concerning histological, genetic, and clinical behavior, thus emphasizing the need for patient-tailored therapy [26]. Cost-effective but less accurate immunohistochemical and in situ hybridization methods allow systematic categorization of breast cancer into estrogen receptor-positive (ER+) luminal, HER2-positive, and triple-negative subtypes, reflecting the type of adjuvant therapy in the clinics [27]. Although molecular assays (Mammaprint^®^, Oncotype DX^®^) predict the outcome of patients with ER+ disease, they are not widespread in hospitals due to the high costs, especially in less developed countries [27], and also add limited information to existing clinical tests [28].

Notably, modern omics studies have allowed a more detailed look at breast cancer categorization and highlighted a few novel targets for its therapy [29,30]. Despite these advances, we lack data concerning the regulation of individual gene expression in the context of known drivers such as *PIK3CA* and *TP53* and more data are needed considering other less recognized or unknown drivers.

Here, we exploited parental taxane-sensitive MCF-7 cells and two taxane-resistant sublines, MCF-7/PacR (named as “PacR” subline) and MCF-7/SB-T-0035R (named as “0035R” sublines), to elucidate mechanisms regulating *TRIP6* expression in breast cancer. We discovered *cis*-acting regulatory motifs in *TRIP6* proximal promoter and clarified its function by dual-luciferase reporter assay. Consequently, we revealed stable hypomethylation in *TRIP6* proximal promoter in tested MCF-7 cells. We found a few *TRIP6* loci localized in der(7)t(7;7)ins(7;15) and one *TRIP6* locus in normal chromosome 7 in MCF-7 cell line. Co-amplified *TRIP6*/*ABCB1* region formed homogeneously stained region inserted in chromosome 3 (PacR subline) and chromosome 19 (0035R subline). Analysis of *TRIP6* expression in 95 breast cancer samples revealed associations of the *TRIP6* mRNA expression level with progesterone receptor positivity and premenopausal status.

## 2. Materials and Methods

### 2.1. Materials

Unless otherwise specified, all chemicals and oligonucleotides were purchased from Merck KGaA (Darmstadt, Germany). Stony Brook Taxane SB-T-0035 was synthesized and kindly provided by Professor Iwao Ojima [31].

### 2.2. Cell Culture

Human breast cancer cell line MCF-7 (RRID: CVCL_0031) was purchased from ATCC. Paclitaxel-resistant MCF-7/PacR (RRID: CVCL_B7P7) (shortened “PacR” in this study) and Stony Brook taxane 0035-resistant MCF-7/SB-T-0035R (RRID: CVCL_C0CU) (shortened “0035R” in this study) sublines were established by multi-step selection on mass populations of MCF-7 cells [32,33,34]. MCF-7 cells were maintained in RPMI-1640 medium supplemented with 10% fetal bovine serum and streptomycin-penicillin mix. Taxane-resistant MCF-7 sublines were cultured in the same medium supplemented with 300 nM paclitaxel (PacR subline) or 300 nM SB-T-0035 (0035R subline). The cell stocks used in this study were independently authenticated (Appendix A).

### 2.3. Collection and Processing of Breast Cancer Tissue Samples

Breast cancer tissue samples (*N* = 95) were collected and snap-frozen during primary surgery in The Faculty Hospital Motol and Institute for the Care for Mother and Child (Prague, Czech Republic) between 2003 and 2009. Sample processing was described in detail previously [35,36]. Samples from 82 patients were collected during the primary surgery before any chemotherapy or hormonal therapy (adjuvant group; ACT group). Samples from the second group of patients (*N* = 13) were collected during the primary surgery after neoadjuvant cytotoxic therapy with regimens containing taxanes or taxanes in combination with 5-fluorouracil and/or anthracycline, and cyclophosphamide (NACT group), a standard regimen in the period of sample collection. Noteworthy, the current guidelines do not support the addition of 5-fluorouracil to the anthracycline (Doxorubicin/Epirubicin)-cyclophosphamide regimen.

A response to NACT was evaluated pre- and post-therapy by ultrasonography. Histological classification of carcinomas was performed according to standard diagnostic procedures [37]. The expression of estrogen and progesterone receptors was assessed immunohistochemically (IHC) with the 1% cut-off value for classification of tumors as hormone receptor positive. *ERBB2* status was defined as positive in samples with IHC score 2+ or 3+ confirmed by fluorescence in situ hybridization or silver in situ hybridization. The cut-off between high and low expression of proliferative marker Ki-67 was 13.25% [38]. Samples were subtyped according to hormone receptor and *ERBB2* expression as triple-negative (TNBC) subtype, *ERBB2* subtype and luminal subtype [39]. Disease-free survival (DFS) was defined as the time elapsed between surgery and disease recurrence.

### 2.4. Isolation of Nucleic Acids and Proteins

Cultured cells were harvested by trypsin-EDTA solution and washed. Breast cancer tissue samples were grounded to powder by mortar and pestle under liquid nitrogen. Nucleic acids and protein were isolated using Allprep DNA/RNA/protein Kit (Qiagen, Hilden, Germany) following the manufacturer’s instructions. Nucleic acids were quantified using Quanti-iT^TM^ PicoGreen^TM^ dsDNA Assay Kit (Invitrogen^TM^, Carlsbad, CA, USA) and Quant-iT RiboGreen RNA Assay Kit (Invitrogen) in Infinite M200 microplate reader (Tecan Group Ltd., Männendorf, Switzerland). RNA integrity was checked by Agilent 2100 Bioanalyzer and Agilent RNA 6000 Nano Assay Kit (Agilent Technologies, Santa Clara, Inc., CA, USA).

### 2.5. Quantitative Reverse Transcription PCR (qRT-PCR)

The real-time PCR study design adhered to the Minimum Information for Publication of Quantitative Real-Time PCR Experiments guidelines [40]. The synthesis of complementary DNA (cDNA) is described in Appendix A. The used TaqMan^®^ Gene Expression probes and PCR conditions are specified in Appendix A. *IPO8* and *MRPL19* were used as reference genes in patient cohorts based on their stability, as previously published [35]. To achieve the best reaction efficiency (>90%), we optimized the cycling conditions of each assay using a calibration curve as described previously [41,42]. For cell lines, the threshold cycle (Ct) of the gene of interest (GOI) was normalized to the reference gene (REF) by the following formula $\Delta Ct=Ct_{REF}-Ct_{GOI\;}$. To compare gene expression between MCF-7 cells and taxane-resistant MCF-7 sublines, we calculated the ΔΔCt value by the following formula $\Delta \Delta Ct=\Delta Ct_{RES}-\Delta Ct_{MCF-7}$. Otherwise, to compare gene expression, fold change was calculated using the 2 ^−ΔΔCt^ method [43].

### 2.6. Assessment of Gene Copy Number

Genomic DNA (10 ng per reaction) was subjected to amplification as triplicate in ABI-PRISM 7500 Fast Real-time PCR System (Thermo Fisher Scientific, Waltham, MA, USA) with Predesigned TaqMan^®^ Copy Number Assays (*TRIP6* FAM-MGB, Cat. No. *Hs02120646_cn; ABCB1* FAM-MGB, Cat. No. *Hs04962504_cn*) and Reference Assays (*TERT* VIC-TAMRA, Cat. No. 4403316; *RPPH1* VIC-TAMRA, Cat. No. *4403326*). The threshold cycle (Ct) value of a gene of interest (GOI) in a sample was normalized to the reference gene (REF) by the following formula $\Delta Ct=Ct_{REF}-Ct_{GOI\;}$. For *ABCB1* in the 0035R subline, data were not corrected by the *TERT* reference gene DNA level as the *TERT* copy number likely altered in these cells (Figure A1). To estimate gene copy number gain or loss in taxane-resistant sublines, we subtracted the normalized (ΔCt) values as follows this formula $\Delta \Delta Ct=\Delta Ct_{RES}-\Delta Ct_{MCF-7\;}$, where RES means taxane-resistant MCF-7 subline.

### 2.7. Western Blot Analysis

Western blot was performed as described elsewhere [44]. The primary antibodies that were used were *anti*-P-glycoprotein (ab3366, RRID: AB_303744), *anti*-actin (ab11003, RRID: AB_297660) from Abcam (Cambridge, UK), and *anti*-TRIP6 (HPA052813, RRID: AB_2681961) from Atlas Antibodies (Bromma, Sweden). SuperSignal™ West Pico PLUS Chemiluminescent Substrate or West Femto Maximum Sensitivity Substrate (Pierce, Rockford, IL, USA) were applied on a membrane to detect the chemiluminescence signal of secondary HRP-conjugated goat *anti*-mouse (SA00001-1, RRID: AB_272565) and HRP-conjugated goat *anti*-rabbit (SA00001-2, RRID: AB_272564) from Proteintech (Rosemont, IL, USA). Images were obtained using ChemiDoc MP imaging system (Biorad, Hercules, CA, USA).

### 2.8. Fluorescence In Situ Hybridization (FISH) Analysis

MCF-7 cells (60% confluency) were incubated with colchicine (0.2 µg/mL) and taxane-resistant MCF-7 sublines were incubated with colchicine and zosuquidar hydrochloride (100 nM) for 1 h. Harvested cells were resuspended in hypotonic buffer (100 mM KCl, 5 mM HEPES, 1 mM EGTA, pH 7.3) and fixed in methanol–glacial acetic acid (3:1). BAC probe mix *ABCB1*(spectrum green)/*TRIP6*(spectrum aqua) was purchased from Empire Genomics (Williamsville, NY, USA) (Appendix A). 24XCyte and XCyte 7 mBAND probes were purchased from MetaSystems (Altlussheim, Germany). All available metaphases were scanned using Metafer AxioImager Z2 – automatic mitoses finder and AxioImager Z1 fluorescence microscope (Carl Zeiss, Oberkochen, Germany) and further analyzed using Isis computer analysis system (MetaSystems). The findings are described according to ISCN2020 [45].

### 2.9. Cloning

pGL3-Promoter, pGL4.10[*luc2*], pGL4.24[*luc2P*/minP], and pNL1.1TK[*Nluc*/TK] vectors were purchased from Promega (Madison, WI, USA). 5′ flanking sequence of the *TRIP6* was taken from Ensembl genome browser (Homo sapiens GRCh38.p12) for *TRIP6*-201 transcript (ENST00000200457.9). Sequence −936/+111 (where +1 means *TRIP6* transcription start) was amplified from MCF-7/PacR genomic DNA by PCR and cloned into pGL3-Promoter vector via *Kpn*I and *Nco*I sites. The inserted sequence was subcloned by PCR into pGL4.10[*luc2*] and pGL4.24 vectors (Appendix A). Mutagenesis of the CRE motif was achieved by cleavage with *Aat*II-HF enzyme followed by 3′ overhangs removal (Large Klenow Fragment, NEB). The construction of plasmids, primers and PCR conditions are summarized in Appendix A. The constructs have been verified by restriction endonuclease cleavage and insert sequencing (LightRun, SupremeRun, Eurofins Genomics, Ebersberg, Germany).

### 2.10. Dual-Luciferase Reporter Assay

Cells (2.0 × 10^5^) were seeded into wells of Nunc^TM^ F96 MicroWell^TM^ plate (Cat. No. 236105, Thermo Fisher Scientific) in paclitaxel-free medium. After 24 h, enabling cells to attach to a surface, the cells were co-transfected with 100 ng of DNA per well at a 100:1 ratio (reporter to co-reporter) using jetPRIME^®^ transfection reagent (Polyplus-Transfection, Illkirch, France) following the manufacturer’s instructions. After 4 h, the transfection mix was replaced by a fresh paclitaxel-free cell culture medium. After 48 h post-transfection, samples were assayed by Nano-Glo^®^ Dual-Luciferase Assay System I (Promega). Plates were read after 10 min of incubation in M200 Pro Plate Reader (Tecan Group Ltd.).

### 2.11. DNA Methylation Profiling

Bisulfite conversion of 500 ng DNA was performed with a EZ DNA Methylation^TM^ Kit (Zymo Research, Irvine, CA, USA), according to the manufacturer’s protocol. The genome-wide DNA methylation was assessed by the Infinium Human MethylationEPIC BeadChip platform (Illumina, San Diego, CA, USA) following the manufacturer’s instructions. The microarray was scanned by the Illumina iScan system. The obtained data were further processed using the R language [46]. Quality control and data normalization were carried out in the *minfi* package as described previously [47,48]. Raw data were converted into β values for further analysis [49,50]. Probes mapped to single nucleotide polymorphism were removed from the analysis [51]. Differentially methylated probes were defined with |Δβ| > 0.2 (20% difference). The β value is defined as the ratio between methylated versus unmethylated alleles.

### 2.12. Bisulfite Sequencing

Extracted DNA (1 µg) was bisulfite converted using the Epitect Bisulfite Kit (Qiagen) following the manufacturer’s instructions. Then, 100 ng of converted DNA was subjected to 42 cycles of amplification (95 °C for 5 min, 95 °C for 30 s, 57 °C for 30 s, 68 °C for 60 s) with Epimark^®^ Hot Start Taq DNA polymerase (NEB), pair of primers (forward: 5′-AGAAATGGTAGTTTAGGGTTTAGGGGGTTA-3′; reverse: 5′-AACCTCTAACCTTCACCCCCTCTTC-3′) in a 50 µL reaction. PCR product was cloned into pGEM^®^ T-Easy vector (Promega). Transformed DH5α Max Efficiency Competent Cells (Thermo Fisher Scientific) were selected on X-Gal/ampicillin plates. Sequencing data were analyzed in Quma online tool [52]. The clones with more than 93% cytosines converted outside CpG were analyzed.

### 2.13. Statistical Analysis

Graphs and statistical analysis were generated in Graph Pad Prism 9.2.0 (GraphPad Software, San Diego, CA, USA) regarding the recommendations described by others [53]. The SPSS v16.0 program (SPSS Inc, Chicago, IL, USA) was used for whole gene CpG methylation data and associations with breast cancer clinic pathological data. The normality of data was tested by the Shapiro–Wilk test prior to statistical analysis. Associations of transcripts with clinical data were assessed by the non-parametric Mann–Whitney, Kruskal–Wallis, and Spearman rank test. All *p*-values were obtained by two-sided tests. A *p*-value of <0.05 was considered statistically significant.

Variances were compared by F-test prior to unpaired *t*-test analysis. The distribution of residuals was checked by residual plot, homoscedasticity plot, and QQ plot. Individual statistical analysis is specified in each figure or table legend.

## 3. Results

### 3.1. TRIP6 as Well as ABCB1 Are Overexpressed in Taxane-Resistant MCF-7 Sublines

Recently, we established Stony Brook 0035-resistant MCF-7 subline (0035R subline) from the same parental MCF-7 cells as paclitaxel-resistant MCF-7 subline (PacR) [34]. SB-T-0035 is a paclitaxel derivative in that a dimethyl carbamoyl group replaces the methyl group at the C10 position of the baccatin core (Figure 1).

We were interested in whether the *TRIP6* gene (7q22.1, 100.8Mb) is overexpressed in 0035R subline similarly to the already described PacR subline [21], and we wanted to identify what mechanisms underpin *TRIP6* overexpression in taxane-resistant MCF-7 sublines. Regarding de novo expression of the adjacent *ABCB1* gene (7q21.12, 87.5Mb), we hypothesized that co-amplification could drive enhanced levels of *TRIP6* and *ABCB1* in PacR cells [33,34].

We compared *TRIP6* and *ABCB1* copy number (Figure 2A), mRNA level (Figure 2B), and protein level (Figure 2C) between MCF-7 cells and taxane-resistant MCF-7 sublines. Due to both the target (*TRIP6*, *ABCB1*) and reference (*RPPH1*, *TERT*) gene copy numbers being unknown in all assayed cell samples, we roughly estimated copy number gain or loss from ΔΔCt values obtained by duplex real-time TaqMan^®^ PCR (Figure 2A, Figure A1). The *ABCB1* and *TRIP6* gene copy number increased in 0035R cells (ΔΔCt*_ABCB1_* = 2.32, ΔΔCt*_TRIP6_* = 2.38) although less than in PacR cells (ΔΔCt*_ABCB1_* = 3.42, ΔΔCt*_TRIP6_* = 3.50). The level of *TRIP6* mRNA increased in 0035R cells (ΔΔCt*_TRIP6_* = 1.35) although less than in PacR cells (ΔΔCt*_TRIP6_* = 2.16) compared to MCF-7 cells. The level of TRIP6 protein increased approximately by 3.5-fold in both taxane-resistant MCF-7 sublines compared to MCF-7 cells (Figure 2B).

Furthermore, we also found markedly elevated levels of *ABCB1* mRNA and protein in 0035R cells, although it was two-fold lower compared to PacR cells (Figure 2B,C).

Collectively, *TRIP6* copy number, mRNA level, and protein level increased in line with *ABCB1* copy number, mRNA, and protein level, suggesting that co-amplification is accountable for their increased expression.

### 3.2. A Few TRIP6 Loci Pre-Exists in MCF-7 Cell Line

Variation in *TRIP6* copy number might underlie high *TRIP6* mRNA and protein expression even in parental MCF-7 cells. In addition, the massive distribution and subcultivation of MCF-7 cells during the last 50 years resulted in enormous MCF-7 cell line heterogeneity [54]. Thus, we first determined the karyotype of MCF-7 cells used in this study.

The composite karyotype of MCF-7 cells assembled from 20 mitoses (of 52 analyzed) counted 67 to 69 chromosomes (Table 1). Chromosomes X, 6, 7, 9, 14, and 21 were disomic. All chromosomes possessed numerical and structural aberrations. Derivative chromosomes were formed predominantly by unbalanced translocations (Figure 3A, Table 1), while reciprocal translocations between t(3;6) and t(4;5) occurred. Chromosome 7 *p*- and q-arm segments translocated to chromosomes X, 2, 7, 10, and 22 (Figure 3B, Figure A2). Subsequent multicolor banding (mBAND) analysis revealed rearrangements in der(7)t(7;7)ins(7;15) (Figure 3B). We detected a few *ABCB1* (7q21.12) and *TRIP6* (7q22.1) loci on both arms of der(7)t(7;7)ins(7;15). Interestingly, the other copy of chromosome 7 was intact and carried *ABCB1* and *TRIP6* loci in situ (Figure 3C).

### 3.3. Chromosome 7 Is Rearranged in Taxane-Resistant MCF-7 Sublines

To validate amplification of the region encompassing *TRIP6* and *ABCB1*, we carried out similar FISH analyses in taxane-resistant MCF-7 sublines.

We karyotyped 8 mitoses (of 34 analyzed) of PacR cells, and 12 mitoses (of 36 analyzed) of 0035R cells (Figure 4, Table 2). The modal chromosome number of taxane-resistant MCF-7 sublines slightly varies from parental MCF-7 cells. Nevertheless, most derivative chromosomes have been preserved (Figure 4, Figure A2). Notably, a breakage at 7q11.2 in the intact chromosome 7 resulted in novel der(7)t(6;7) and der(7)del(7)(p12)del(7)(q11.1) in taxane-resistant MCF-7 sublines (Figure 4, Figure A3). We detected *TRIP6* co-amplification with *ABCB1* (Figure 4C,F) as a homogeneously stained region (HSR) translocated to chromosome 3 (PacR subline) or chromosome 19 (0035R subline). In a few mitoses of 0035R cells, we unambiguously detected HSR in chromosome 15 (Figure 4D), indicating that 0035R subline might consist of two subclones.

### 3.4. TRIP6 Expression Is Regulated by Cyclic AMP Response Element (CRE)

*TRIP6* promoter has not been functionally characterized yet. Hence, we generated 5′ and 3′ truncated *TRIP6* promoter reporter constructs by cloning a human *TRIP6* promoter sequence upstream of the minimal synthetic promoter (minP) in pGL4.24[*luc2P*/minP] vector (Appendix A). In fact, the cloned full-length *TRIP6* promoter sequence encompassed *SLC12A9* exon 14 (sequence −936 to −376, relative to *TRIP6* transcription start, TSS) and *SLC12A9-TRIP6* intergenic region (sequence −375 to −1) (Figure 5).

We assessed firefly (Fluc2P) and deep-sea shrimp Nanoluc (Nluc) luciferase activities in MCF-7 cells and taxane-resistant MCF-7 sublines co-transfected with a series of 5′ and 3′ truncated constructs and the normalization pNL1.1.TK[Nluc/TK] vector. Firstly, the experiments showed that the *TRIP6* proximal promoter (sequence −157 to −12 relative to the *TRIP6* transcription start) but not the *TRIP6* distal promoter (sequence −936 to −157) is sufficient to drive *TRIP6* expression in both MCF-7 cells and taxane-resistant MCF-7 sublines (Figure 5). Secondly, the construct −72/−12 achieved significant Fluc2P/Nluc activity (20%, 28%, and 37% of relative Fluc2P/Nluc activity of the −157/−12 construct in MCF-7, PacR, and 0035R cells, respectively).

To identify *cis*-acting regulatory elements in the active human *TRIP6* proximal promoter, we scanned the −200 to −1 sequence with Jaspar 2022 transcription factor binding profiles (≥93% relative profile score threshold) (Appendix A) [55]. We manually identified core elements within most of the predicted binding sites (Figure 6A). Remarkably, we discovered a full cyclic AMP response element (CRE) motif at position −60 to −53, corresponding to the −72/−12 construct with marked activity. Mutagenesis of CRE demonstrated a 6- to 21-fold reduction in Fluc2P/Nluc activity in 5′ truncated constructs (−157/−12ΔCRE, −117/−12ΔCRE, and −72/−12ΔCRE) (Figure 6B), indicating that CRE is crucial to *TRIP6* transcription in MCF-7 cells.

Furthermore, the −117/−72 construct exhibited a two-fold increase in Fluc2P/Nluc activity compared to the −72 /−12 construct in all tested cells (Figure 6B). The region −117 to −72 encompasses an enhancer box (E box) and CT box [56]. Furthermore, activating protein 1 (AP-1) motif located within the region −157 to −117 weakly stimulated (1.6-fold, *p* = 0.036) Fluc2P/Nluc activity in MCF-7 cells but not in PacR and 0035R cells (1.2-fold, *p* = 0.60, 1.1-fold, *p* = 0.33, respectively). Yet, the other element(s), probably the GC box [57] or M-CAT [58], increased basal expression as seen in the −72/−12ΔCRE construct. We recurrently detected no stimulatory activity in the region −45 to −12 in all tested cells.

Collectively, CRE unambiguously promotes *TRIP6* transcription in MCF-7 cells and taxane-resistant MCF-7 sublines. The predicted E box, GC box, CT box, and M-CAT might contribute to the *TRIP6* promoter activity; however, there would still be other unidentified motifs. In addition, the AP1 site likely enhances *TRIP6* transcription only in MCF-7 cells, as it does not appear to modulate the response in PacR and 0035R cells.

### 3.5. TRIP6 Proximal Promoter Is Hypomethylated in Taxane-Resistant MCF-7 Sublines

Methylation of CpG site in CRE motif hampers transcription in *cis* [59]. Considering *TRIP6* dependence on the CRE motif (Figure 6), we assessed the methylation of 8 CpG sites within the *TRIP6* proximal promoter by bisulfite PCR. As it is shown in Figure 7A, the analyzed region exhibits hypomethylation in MCF-7 cells (3.8%), PacR subline (6.0 %), and 0035R subline. Importantly, we detected an unmethylated CpG in the CRE motif in all tested cells (Figure 7A).

Furthermore, we assessed *TRIP6* methylation in clinical breast cancer samples (TCGA study) (Figure 7B). *TRIP6* mRNA expression negatively correlated (Spearman’s coefficient = −0.52, *p* < 0.001) with CpG methylation level, indicating that DNA methylation might regulate *TRIP6* expression also in breast tumors.

To explore *TRIP6* differential methylation between MCF-7 cells and taxane-resistant MCF-7 sublines (PacR, 0035R), we employed 16 probes that targeted to defined gene regions TSS200 (−200 bases to TSS, i.e., proximal promoter), TSS1500 (−1500 to −200, i.e., distal promoter), 1st Exon, gene body (region between ATG and stop codon), and 5′ and 3′ untranslated regions (UTRs). It is worth noting that the distal promoter region substantially overlaps with the last exon of the *SLC12A9* gene. Although the whole *TRIP6* gene sequence analysis showed higher methylation in PacR cells (*p* = 0.004, FDR = 0.025), in fact, methylation of the distal *TRIP6* promoter region (TSS1500) significantly changed. We found no differential methylation between MCF-7 cells and taxane-resistant MCF-7 sublines in the *TRIP6* proximal promoter (TSS200), gene body, and 1st exon, in line with bisulfite sequencing data (Table 3).

These findings suggest that the *TRIP6* proximal promoter is stably hypomethylated, thereby contributing to high *TRIP6* expression in MCF-7 cells and taxane-resistant MCF-7 sublines.

### 3.6. Associations of TRIP6 mRNA Level with Clinicopathological Features of Breast Cancer

In a recent study, Zhao et al. postulated *TRIP6* as a putative prognostic biomarker in breast cancer [20]. Therefore, we aimed to validate this finding by evaluating *TRIP6* expression in 95 breast tumor tissue samples and 6 non-tumor tissues collected in the Czech Republic.

Table 4 summarizes clinical data, response to the therapy, and survival of patients who provided breast cancer tissues. The median age (± SD) of patients with a breast cancer diagnosis was 56.0 ± 10.7 years. Most individuals were diagnosed with invasive ductal carcinoma (84.2%), grade 1 or 2 (75.8%), and stage II (62.1%). Nearly all breast cancer tissues expressed estrogen receptor (ER, 90.5% of samples) and progesterone receptor (PR, 70% of samples), meaning that luminal molecular subtype (91.6%) prevailed in evaluated samples. The median of disease-free survival (DFS) (± SD) of patients was 61.1 ± 28.4 months, and overall survival was 70.9 ± 28 months. Unfortunately, disease progression occurred in 9 of the 95 patients, and 8 patients died.

We assessed *TRIP6* mRNA expression in all collected breast tissue samples (*N* = 95) and protein expression only in a small number of samples (*N* = 20) due to limitations in sample size. Whereas all breast tumor tissues expressed *TRIP6* mRNA, we detected TRIP6 protein expression by immunoblotting in 17 of the 20 examined samples (Figure 8A). *TRIP6* mRNA and protein level correlated intermediately (Spearman’s coefficient 0.594, *p* = 0.032) in breast cancer tissues. (Figure 8B).

We found no difference in *TRIP6* mRNA expression levels between adjuvant (*N* = 82) and neoadjuvant (*N* = 13) cohorts (*p* = 0.86). Furthermore, we found no statistically significant correlation between *TRIP6* mRNA level and DFS or OS, independent of the type of therapy. High *TRIP6* mRNA expression was observed in premenopausal (*p* = 0.033) and progesterone receptor positive (*p* = 0.020) breast cancer in the adjuvant cohort of breast cancer patients but not in the neoadjuvant cohort (*p* = 0.50 and *p* = 0.77, respectively) (Table 5).

## 4. Discussion

An early study demonstrated a ubiquitous 1.8-Kb *TRIP6* mRNA expression in human organs except for skeletal muscle, brain, and leukocytes [60]. Recently, Shukla et al. detected TRIP6 in ependymal and choroid plexus cells of embryonic and early post-natal (to P10) mice brains [12]. Additionally, observations of enhanced *TRIP6* expression in various neoplasms might indicate disrupted gene regulatory mechanisms during cancerogenesis [20,61]. Furthermore, we found *TRIP6* overexpression in paclitaxel-resistant MCF-7/PacR subline [62], yet the molecular mechanism(s) that drive *TRIP6* expression in MCF-7 cells as well in paclitaxel-resistant cells have not been described in detail.

Herein, we revealed that *TRIP6* copy number gain and the activity of the cyclic-AMP response element in the hypomethylated *TRIP6* proximal promoter contribute to the high TRIP6 protein level in parental MCF-7 cells. Although the AP-1 motif seems more important in parental MCF-7 cells, copy number gain but not altered regulation of the *TRIP6* promoter instead contribute to *TRIP6* overexpression in both taxane-resistant MCF-7 sublines (Figure 9).

While *TRIP6* mRNA levels differed between PacR cells and 0035R cells, TRIP6 protein abundance was identical and markedly higher compared to parental MCF-7 cells (Figure 2). This discrepancy might indicate the differential *TRIP6* post-transcription regulation, for instance, by putative differential *TRIP6* mRNA base modifications, miRNA, or recently observed TRIP6 ubiquitin-mediated degradation [24]. Concerning miRNA, it was reported that miR-7, miR-138-5p, miR-485-3p, and miR-589-5p regulate *TRIP6* gene expression; unfortunately, their function related to *TRIP6* in breast cancer has not been investigated [21,22,23,63]. However, our preliminary data suggest that miR-138-5p is not expressed in MCF-7 cells and taxane-resistant MCF-7 sublines (personal communication Dr. R. Václavíková).

Strikingly, the *TRIP6* (100.8 Mb, 7q22.1) gene copy number and mRNA level increased in parallel with the *ABCB1* (87.5 Mb, 7q21.12) gene copy number and mRNA level. In agreement, FISH analyses unambiguously validate *TRIP6*/*ABCB1* co-amplification in taxane-resistant MCF-7 sublines (Figure 4). Despite *TRIP6* amplification, our findings indicate that TRIP6 is not involved in resistance to taxanes, as silencing of the *TRIP6* does not seem to affect the response of the 0035R cells to SB-T-0035 compound (Appendix A). To date, upregulation of the *TRIP6* gene occurred in daunorubicin- (EPG85-257RDB) and mitoxantrone-resistant (EPG85-257RNOV) human gastric carcinoma cells, the former cells having also upregulated *ABCB1* [64].

The *ABCB1* amplified region, referred to as *ABCB1* amplicon, is often documented in drug-resistant sublines [64,65,66,67]. By retrospective analysis, Genovese et al. defined the *ABCB1* amplicon core as a 1 Mb region commonly detected in ABCB1 overexpressing cells [65]. By contrast, the largest reported *ABCB1* amplicon was bordered by semaphorin 3D (*SEMA3D*, 84.3 Mb) and cyclin-dependent kinase 6 (*CDK6*, 92.1 Mb) [66]. The occurrence of *TRIP6* (7q22.1, 100.8Mb) might indicate the extraordinary size of the *ABCB1* amplicon in MCF-7 sublines.

The most famous breakage–fusion–break (BFB) mechanism of amplicon formation leverages specific sequences referred to as fragile sites [68]. What mechanism specifies fragile site selection is not well known. The breaks likely occurred at the FRA7F aphidicolin site (98.7–107.4 Mb) and the 7q11.2 region. The order of events is challenging due to the utilization of multiple selection steps and no direct observation of fusion bridges. Nevertheless, numerical aberration of chromosome 7 in taxane-resistant MCF-7 sublines might be a remnant of dicentric chromosome 7.

Beyond *ABCB1*/*TRIP6* co-amplification, we noticed the loss of der(18)t(18;22) in both taxane-resistant MCF-7 sublines. The impact of this aberration in the context of taxane resistance is unknown.

Since the enhanced *TRIP6* expression could theoretically be caused by different transcriptional regulations at the *TRIP6* promoter level, we analyzed the responsiveness of the *TRIP6* promoter by dual-luciferase assay. The *TRIP6* proximal promoter region (−157 to −45) controlled luciferase expression in MCF-7 cells (Figure 5). The most active segment spanning −72 to −45 nucleotides harbors the M-CAT motif, cyclic AMP response element (CRE), and GC box (Figure 6). Disrupting the CRE motif by mutagenesis reduced luciferase activity, highlighting its pivotal role in *TRIP6* transcription regulation (Figure 6). A genome-wide analysis has previously identified identical CRE motif within the *TRIP6* proximal promoter [69], but its role has remained elusive. The *cis*-regulating activity of the CRE motif relies on its position (< 250 bases) relative to gene transcription start [70] and methylation of the inner CpG site [59]. As tested by bisulfite sequencing (Figure 7A), the CpG site within the CRE motif was not methylated in MCF-7 cells and in both taxane-resistant cells; however, whether this particular methylation affects the expression of *TRIP6* remains to be determined in further studies.

Finally, we evaluated the clinical data of breast cancer patients with *TRIP6* mRNA expression. Recently, we revealed no clinicopathological association of the *TRIP6* mRNA expression level in ovarian cancer [71]. To highlight our findings concerning the regulation of *TRIP6* expression in sensitive and taxane-resistant MCF-7 breast cancer cell lines, we evaluated *TRIP6* mRNA expression against clinical data of breast cancer patients who had undergone taxane-containing regimens. So far, Zhao et al. have analyzed TRIP6 protein expression in breast cancer from the Chinese cohort [20]. Unfortunately, our data did not validate most of the published results, likely due to the small number of patients in our study and the heterogenous nature of breast cancer.

## 5. Conclusions

This study presents compelling evidence that the cyclic AMP response element (CRE) located within the stable hypomethylated proximal promoter controls *TRIP6* expression in MCF-7 cells. Furthermore, increased *TRIP6* copy number contributes to high TRIP6 expression in MCF-7 cells in vitro. Co-amplification of *TRIP6* with *ABCB1* underlies *TRIP6* upregulation in two taxane-resistant MCF-7 sublines. Cytogenetic analyses showed that amplicon arose from intact chromosome 7. In addition, we observed a loss of derivative chromosome der(18)t(18;22) in both sublines, with an unknown relation to taxane resistance. Moreover, the present study has not found direct prognostic or predictive relevance of *TRIP6* for better tailoring breast cancer management at the clinics. Instead, the analysis of breast tumor of a neoadjuvant cohort revealed *TRIP6* mRNA expression level associations with positive progesterone receptor expression status and premenopausal status.

Collectively, we propose that *TRIP6* proximal promoter might act as another important regulatory site in regulation of *TRIP6* expression. The relevance of our functionally valid observation for clinical course of breast and other cancer(s), including eventual utility of TRIP6 as a target for new therapy design, shall be evaluated by follow-up studies.

## Figures and Tables

**Figure 1 genes-14-00296-f001:**
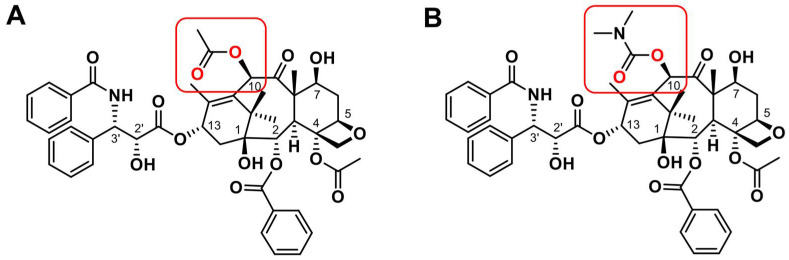
Structural formulas of taxanes used in the study. (**A**) Paclitaxel; (**B**) Stony Brook Taxane 0035 (SB-T-0035). Substituents bound at the C10 position of the baccatin III core are highlighted by the red frame.

**Figure 2 genes-14-00296-f002:**
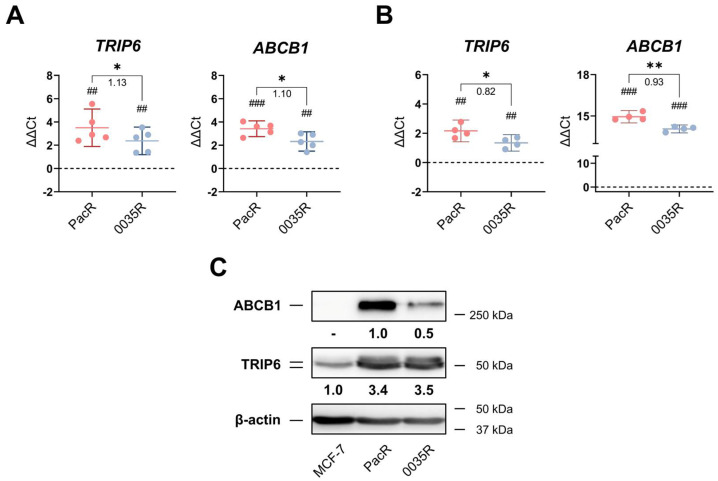
*TRIP6* and *ABCB1* are overexpressed in paclitaxel-resistant MCF-7 subline (PacR) and SB-T-0035-resistant MCF-7 (0035R) subline. (**A**) The DNA level of *TRIP6* and *ABCB1* plotted as ΔΔCt values (*N* = 5, 3 technical replicates). The dashed line represents the DNA level in parental MCF-7 cells. For *ABCB1* in the 0035R subline, data were not corrected by the *TERT* reference gene DNA level. The numbers below the zig–zag lines represent DNA level difference between PacR and 0035R sublines. Statistical analysis was performed using the one-sample *t*-test (#) and paired *t*-test (*). (**B**) *TRIP6* and *ABCB1* mRNA expression level was plotted as delta-delta threshold cycle (ΔΔCt) values (*N* = 4, 3 technical replicates). Dashed lines represent *TRIP6* or *ABCB1* expression in MCF-7 cells. The mean and 95% confidence interval (CI) are shown. The numbers below the zig–zag lines represent the mRNA expression level difference between PacR and 0035R sublines. Statistical significance was tested using the one-sample *t*-test (#) and paired *t*-test (*). (**C**) Western blot analysis of ABCB1 transporter and TRIP6 in MCF-7 cells and taxane-resistant MCF-7 sublines. The numbers displayed below each representative Western blot mean the fold change of band volume normalized to β-actin level (*N* = 4); (**D**) * *p* < 0.05, ** *p* < 0.01. ## *p* < 0.01, ### *p* < 0.001.

**Figure 3 genes-14-00296-f003:**
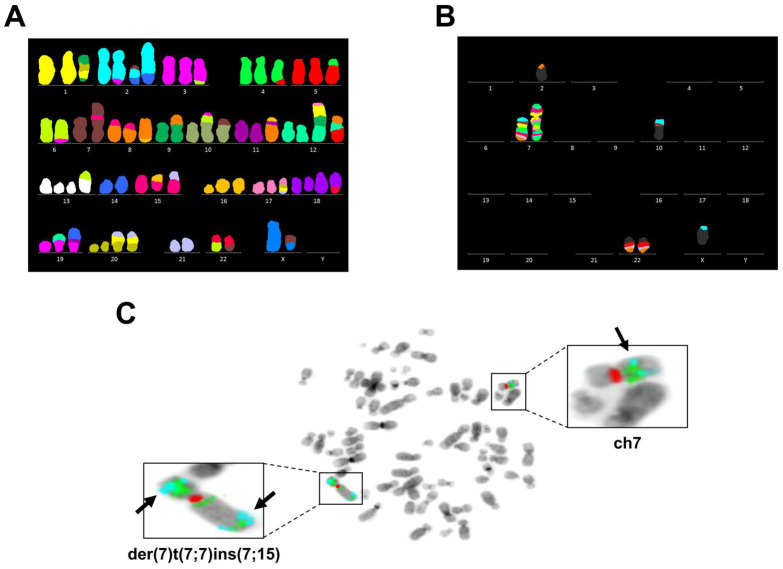
*TRIP6* and *ABCB1* possess increased copy numbers in parental MCF-7 cell line. (**A**) Multicolor fluorescence in situ hybridization (FISH) analysis of representative metaphase chromosomes of MCF-7 cells. (**B**) Multicolor banding (mBAND) of chromosome 7 analysis of representative metaphase chromosomes of MCF-7 cells. (**C**) Dual FISH analysis of representative metaphase chromosomes of MCF-7 cells. Labeled chromosomes are shown in detail. Arrows point to *TRIP6* loci. Detected signals come from a probe specific to *ABCB1* (green), *TRIP6* (aqua), and the centromere of chromosome 7 (red).

**Figure 4 genes-14-00296-f004:**
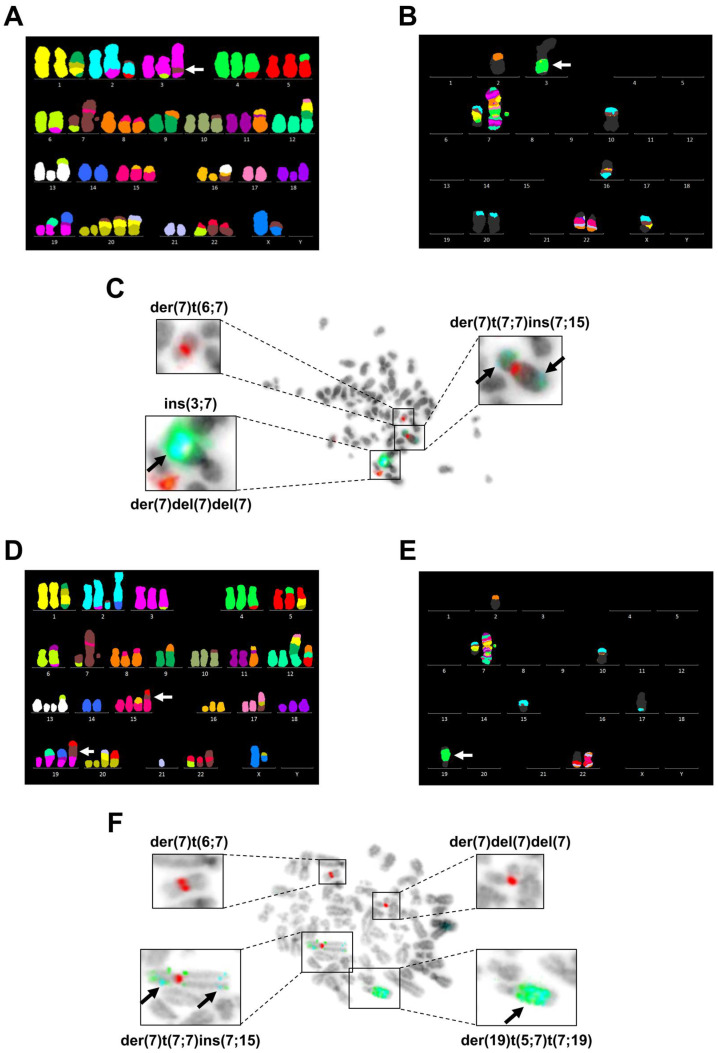
Co-amplification of *TRIP6* and *ABCB1* genes in taxane-resistant MCF-7 sublines (PacR, 0035R). Representative metaphase chromosomes of PacR subline analyzed by (**A**) multicolor fluorescence in situ hybridization (mFISH), (**B**) multicolor band (mBAND) of chromosome 7, and (**C**) dual FISH. Representative metaphase chromosome of 0035R subline analyzed by (**D**) mFISH, (**E**) mBAND of chromosome 7, and (**F**) dual FISH.The white arrows point at homogeneously staining regions (HSRs) harboring the *ABCB1* and *TRIP6* genes. Labeled chromosomes are shown in detail. Detected signals come from a probe specific to *ABCB1* (green), *TRIP6* (aqua), and the centromere of chromosome 7 (red).

**Figure 5 genes-14-00296-f005:**
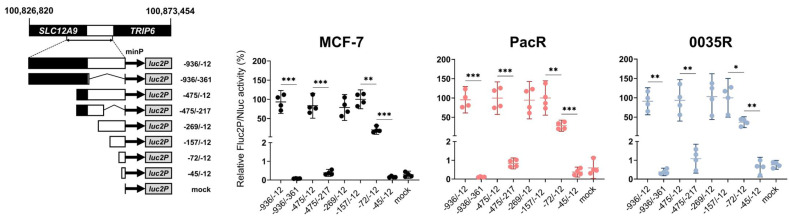
The *TRIP6* proximal promoter governs *TRIP6* transcription in parental MCF-7 cell line as well as in taxane-resistant MCF-7 sublines (PacR, 0035R). Schematic diagrams of the 5′ and 3′ truncated constructs (on the left) and scatter dot plots showing normalized luciferase activities (Fluc2P/Nluc) relative to the construct −157/−12 (on the right). The cloned sequence −936/−12 encompasses an intergenic region (white box, sequence −12 to −375) and a part of the *SLC12A9* upstream gene (black box, sequence −376 to −936). minP refers to synthetic minimal TATA box promoter. The mean and 95% confidence interval (CI) are displayed for each construct (*N* = 4, 3 technical replicates). Empty vector pGL4.24[*luc2P*/minP] served as a mock. Statistical significance was tested using the one-way blocked ANOVA with Geisser–Greenhouse correction followed by Tukey’s post hoc test on log-transformed data. * *p* < 0.05, ** *p* < 0.01, *** *p* < 0.001.

**Figure 6 genes-14-00296-f006:**
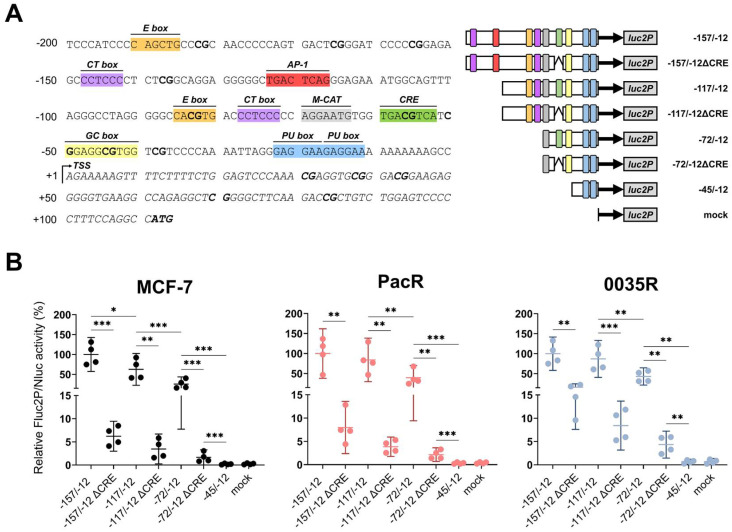
Cyclic AMP response element (CRE) regulates *TRIP6* proximal promoter activity in parental MCF-7 cell line as well as in taxane-resistant MCF-7 sublines (PacR, 0035R). (**A**) Predicted *cis*-regulatory elements in the human *TRIP6* proximal promoter sequence (GRCh38.p12) (on the left). The sequence also includes a 5′ untranslated region and *TRIP6* start codon. CpG dinucleotides and start codon are highlighted in bold. TSS means *TRIP6* transcription start. E box refers to enhancer box, CT box refers to CT-rich sequence, AP-1 site refers to Activator protein 1, M-CAT refers to muscle-CAT element (in reverse orientation), CRE refers to cyclic AMP response element, GC-box refers to GC-rich sequence. Schematic diagrams of the *TRIP6* proximal promoter with wild-type CRE or mutated CRE motif (on the right). Colored rectangles correspond to predicted *cis*-acting gene regulatory elements positioned at the 5′ flanking sequence of *TRIP6*. (**B**) Scatter dot plots showing Fluc2P/Nluc activities relative to the −157/−12 construct (on the right). minP refers to synthetic minimal TATA box promoter. The mean and 95% confidence interval (CI) are displayed for each construct (*N* = 4, 3 technical replicates). Empty vector pGL4.24[*luc2P*/minP] served as a mock. Statistical significance was tested using the one-way blocked ANOVA with Geisser–Greenhouse correction followed by Tukey’s post hoc test on log-transformed data. * *p* < 0.05, ** *p* < 0.01, *** *p* < 0.001.

**Figure 7 genes-14-00296-f007:**
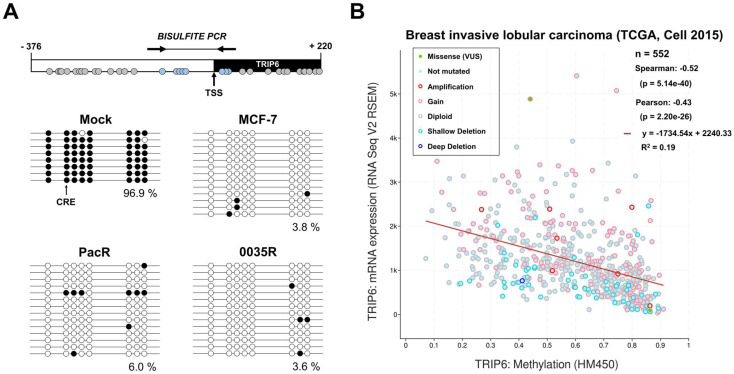
The *TRIP6* proximal promoter is stable hypomethylated in parental MCF-7 cell line as well as in taxane-resistant MCF-7 sublines (PacR, 0035R); (**A**) Schematic diagram (top) displaying the position of individual CpG dinucleotides (grey circles) in *TRIP6*-*SLC12A9* intergenic region (white rectangle, −376 to −1) and *TRIP6* exon 1 (black rectangle, +1 to +220). Methylation status of CpG sites estimated by bisulfite sequencing (bottom). In vitro methylated human diploid DNA served as an internal control (mock). The black arrow points CpG dinucleotide within the CRE motif. The white circles depict non-methylated CpG dinucleotides. Black circles depict methylated CpG dinucleotides. Analyzed CpG dinucleotides are shown as blue circles. Arrows indicate the position of the used primer pair. TSS means *TRIP6* transcription start. (**B**) Correlation of *TRIP6* mRNA level (normalized to RNA-Seq by Expectation–Maximization method, RSEM) with *TRIP6* methylation (HM450 Illumina Platform) in The Cancer Genome Atlas (TCGA) breast cancer study (*N* = 552). VUS refers to a variant of unknown significance. Shallow deletion refers to possible heterozygous deletion. Deep deletion refers to possible homozygous deletion. The graph was generated in the cBioPortal platform.

**Figure 8 genes-14-00296-f008:**
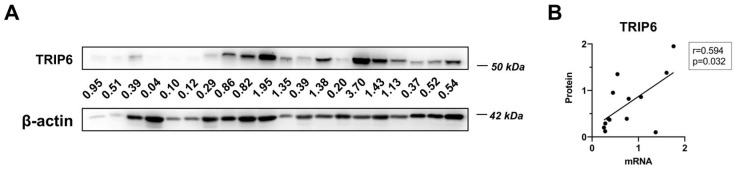
*TRIP6* expression in breast cancer samples. (**A**) TRIP6 protein expression in 20 breast tumor samples. The numbers displayed below the TRIP6 Western blot represent the TRIP6 level normalized to the β-actin level. (**B**) Correlation of *TRIP6* mRNA level versus TRIP6 protein in 13 breast cancer samples. Spearman’s correlation coefficient and statistical significance (*p*-value) are shown.

**Figure 9 genes-14-00296-f009:**
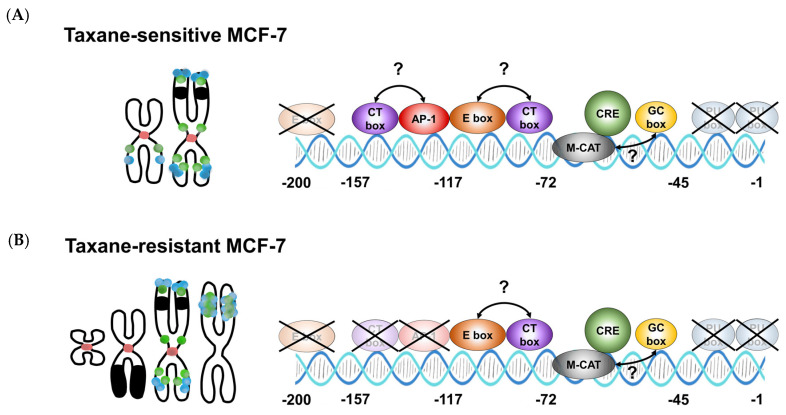
Chromosomal aberrations and *cis*-regulatory elements in the TRIP6 promoter contribute to enhanced TRIP6 expression in parental MCF-7 cells and taxane-resistant MCF-7 sublines. (**A**) In MCF-7 cells, copy number gain (on the left) and the activity of the cyclic AMP response element (on the right) in the hypomethylated TRIP6 proximal promoter contribute to high TRIP6 expression. The role of other putative transcription factor binding sites remains elusive. (**B**) In MCF-7 sublines, TRIP6/ABCB1 co-amplification led to the formation of a homogeneously stained region in chromosome 3 (PacR) or 19 (0035R) (on the left). In contrast, the activity of elements in the hypomethylated TRIP6 promoter remained unchanged except for the region containing the putative AP-1 site (on the right). Dots represent TRIP6 loci (blue) and ABCB1 loci (green). The red area refers to chromosome 7 centromere. The black area refers to chromosome 15 (in MCF-7 and taxane-resistant MCF-7 cells) or chromosome 6 segments (in taxane-resistant MCF-7 cells only).

**Table 1 genes-14-00296-t001:** Composite karyotype of parental MCF-7 cell line. Differences to taxane-resistant MCF-7 sublines are highlighted in bold.

Cell Line	Composite Karyotype
MCF-7 67~69 <3n>	X,-X,der(X)t(X;7)(p11.2;p15.3)del(X)(q21.3q28),der(1)t(9;20)(?;?)t(1;20)(p12;?)t(1;9)(q21;?),+2,der(2)t(2;3)(q?34;?),der(2)t(2;7)(p?;q32)t(2;14)(q36;q?),der(2)t(2;14)(q36;q?),der(3)del(3)(p?13p?26)t(3;6)(q27;q25),t(4;5)(q?31;p13),-6,der(6)t(3;6)(q27;q25),**-7**,der(7)t(7;7)(p15;?)ins(7;15)(p;?),der(8)t(8;15)(p11.2;q?)x2,**der(8)t(8;16)(q?24;?)**,-9,der(9)t(8;9)(q?;p22),**der(10)t(3;6)(?;?)t(6;10)(?;p14)**,der(10)t(7;10)(p21,p14),der(11)t(11;16)(p11.2;?)t(8;11)(q11.2;q13),+12,der(12)t(1;17)(?;?)t(1;9)(?;?)t(9;12)(?;p11.2),der(12)t(8;12)(?;p11.2)t(5;12)(?;q?21),+13,del(13)(q?22q?34)x2,der(13)t(6;13)(?;p11.2),-14,+15,der(15)t(15;16)(p11.2;q?)del(15)(q?22.3q?26),der(15)t(15;21)(p11.2;q?21),der(16)del(16)(p?13)del(16)(q?21),**der(17)t(17;20)(q?24;?)t(1;20)(?;?)t(1;21)(?;?),+18**,der(18)del(18)(p?11.2)del(18)(q?11.2),dup(18)(q?q?),**der(18)t(18;22)(p11.2;q11.2)**,der(19)t(12;19)(q13;p13.3),der(19)t(11;14)(?;q?)t(11;19)(?;p12),+20,der(20)t(1;21)(?;q?)t(1;20)(?;p11.2)x2,-21,der(22)t(6;22)(?;q11.2),der(22)t(7;22)(q22;q11.2)x2[cp20]

**Table 2 genes-14-00296-t002:** Composite karyotype of taxane-resistant MCF-7 sublines. Differences to parental MCF-7 cell line are highlighted in bold.

Cell Subline	Composite Karyotype
PacR 67-68 <3n>	X,-X,der(X)t(X;7)(p11.2;p15.3)del(X)(q21.3q28),der(1)t(9;20)(?;?)t(1;20)(p12;?)t(1;9)(q21;?),+2,der(2)t(2;3)(q?34;?),**der(2)t(2;7)(p?;q32)t(2;5)(q?;?)**,der(2)t(2;14)(q36;q?),der(3)del(3)(p?13p?26)t(3;6)(q27;q25),**ins(3;7)(q?23;q11.2q22)**,t(4;5)(q?31;p13),-6,der(6)t(3;6)(q27;q25),**der(7)t(6;7)(?;q11.2)**,der(7)t(7;7)(p15;?)ins(7;15)(p;?),**der(7)del(7)(p12)del(7)(q11.1)**,der(8)t(8;15)(p11.2;q?)x2,-9,der(9)t(8;9)(q?;p22),der(10)t(7;10)(p21;p14),der(11)t(11;16)(p11.2;?)t(8;11)(q11.2;q13),+12,der(12)t(1;17)(?;?)t(1;9)(?;?)t(9;12)(?;p11.2),der(12)t(8;12)(?;p11.2)t(5;12)(?;q?21),del(13)(q?22q?34),der(13)t(6;13)(?;p11.2),-14,**der(15)t(15;16)(p11.2;q?)**,der(15)t(15;16)(p11.2;q?)del(15)(q?22.3q?26),**+16**,der(16)del(16)(p?13)del(16)(q?21)x2,**der(16)t(13;16)(q?12;p11.2)t(7;16)(p15.3;q?13),-17**,der(18)del(18)(p?11.2)del(18)(q?11.2),dup(18)(q?q?),der(19)t(12;19)(q13;p13.3),der(19)t(11;14)(?;q?)t(11;19)(?;p12),**+20x2,der(20)t(1;7)(?;p21)t(1;20)(?;p11.2)x2**,der(20)t(1;21)(?;q?)t(1;20)(?;p11.2),-21,der(22)t(6;22)(?;q11.2),der(22)t(7;22)(q22;q11.2)x2[cp8]
0035R 67~70 <3n>	X,-X,**der(X)t(X;20)(p11.2;?)del(X)(q21.3q28)**,der(1)t(9;20)(?;?)t(1;20)(p12;?)t(1;9)(q21;?),+2,der(2)t(2;3)(q?34;?),**der(2)t(2;7)(p?12;q32)del(2)(q36)**,der(2)t(2;14)(q36;q?),der(3)del(3)(p?13p?26)t(3;6)(q27;q25),t(4;5)(q?31;p13),**der(5)t(1;5)(q?13)t(1;20)(?;q?)**,-6,**der(6)t(6;11)(p?21;q?)t(3;6)(q27;q25),der(7)t(6;7)(?;q11.2)**,der(7)t(7;7)(p15;?)ins(7;15)(p;?),**der(7)del(7)(p12)del(7)(q11.1)**,der(8)t(8;15)(p11.2;q?)x2,-9,der(9)t(8;9)(q?;p22),der(10)t(7;10)(p21;p14),der(11)t(11;16)(p11.2;?)t(8;11)(q11.2;q13),+12,der(12)t(1;17)(?;?)t(1;9)(?;?)t(9;12)(?;p11.2),der(12)t(8;12)(?;p11.2)t(5;12)(?;q?21),+13,del(13)(q?22q?34)x2,der(13)t(6;13)(?;p11.2),-14,+15,der(15)t(15;16)(p11.2;q?)del(15)(q?22.3q?26),**der(15)t(5;7)(?;p22)t(7;15)(p22;p11.2)**,der(16)del(16)(p?13)del(16)(q?21),**der(17)t(17;20)(q?24;?)t(7;20)(p22;?)**,der(18)del(18)(p?11.2)del(18)(q?11.2),dup(18)(q?q?),**+19**,der(19)t(12;19)(q13;p13.3),der(19)t(11;14)(?;q?)t(11;19)(?;p12),**der(19)t(5;7)(?;q21)t(7;19)(q21;p13.3)**,+20,der(20)t(1;21)(?;q?)t(1;20)(?;p11.2),**der(20)t(5;20)(?;p?12)**,-21,der(22)t(6;22)(?;q11.2),der(22)t(7;22)(q22;q11.2),**der(22)t(7;22)(q22;p11.2)t(7;22)(q22;q11.2)**[cp12]

**Table 3 genes-14-00296-t003:** Altered *TRIP6* methylation level between parental MCF-7 cells and taxane-resistant MCF-7 sublines (PacR, 0035R). The absolute β value (|Δβ|) is defined as difference between β value of MCF-7 cells and PacR cells, or MCF-7 cells and 0035R cells. Statistical analysis results include *p*-value and false discovery rate (FDR). Ns means statistically insignificant result.

Region *	MCF-7 vs. PacR			MCF-7 vs. 0035R		
	|Δβ|	*p*-Value	FDR	|Δβ|	*p*-Value	FDR
Whole gene	0.41	0.004	0.025	0.36	0.2	ns
1st exon	0.23	0.121	ns	0.01	0.439	ns
Body	0.12	0.248	ns	0.11	0.248	ns
TSS1500	0.49	0.02	0.025	0.36	0.002	0.008
TSS200	0.40	0.05	ns	0.05	0.05	ns

* Gene regions as defined in Results section.

**Table 4 genes-14-00296-t004:** Clinicopathological characteristics of 95 breast carcinoma patients.

Characteristics	Breast Carcinoma Set
**Mean age at diagnosis, years**	56.0 ± 10.7
**Menopausal status**	**N (%)**
Premenopausal	27 (28.4)
Postmenopausal	66 (69.5)
Not available	2 (2.1)
**Histological type**	**N (%)**
Invasive ductal carcinoma	80 (84.2)
Others	15 (15.8)
**Histological grade**	**N (%)**
G1	13 (13.7)
G2	59 (62.1)
G3	22 (23.1)
Not available	1 (1.1)
**Stage**	**N (%)**
I	31 (32.6)
II	59 (62.1)
III	4 (4.2)
IV	1 (1.1)
**Estrogen receptor status**	**N (%)**
Positive	86 (90.5)
Negative	9 (9.5)
**Progesterone receptor status**	**N (%)**
Positive	70 (73.7)
Negative	25 (26.3)
**ERBB2 status**	**N (%)**
Positive	27 (28.4)
Negative	68 (71.6)
**Ki67 status** ^1^	**N (%)**
Positive	70 (73.7)
Negative	13 (13.7)
Unknown	12 (12.6)
**Molecular subtype**	**N (%)**
Luminal A	60 (63.2)
Luminal B	26 (27.3)
ERBB2	7 (7.4)
Triple negative	2 (2.1)
**Therapeutic regimens**	**N (%)**
Neoadjuvant (NACT) ^2^	13 (13.7)
Adjuvant (ACT)	82 (86.3)
**Relapse**	**N (%)**
Yes	9 (9.5)
No	86 (90.5)
**Overall survival (OS)**	
Mean (months) ± SD	70.9 ± 28.0
**Disease-free survival (DFS)**	
Mean (months) ± SD	61.1 ± 28.4

^1^ The cut-off score was 13.25% [38]. ^2^ Including the individual in stage IV and undergoing a palliative care.

**Table 5 genes-14-00296-t005:** Significant associations of intratumoral *TRIP6* mRNA level with clinical data of breast carcinoma patients in the adjuvant chemotherapy group (*N* = 82).

Characteristics	*TRIP6* Expression Relative to *IPO8* and *MRPL19*	Significance (Mann–Whitney)
Premenopausal	1.05 ± 0.59	0.033
Postmenopausal	0.77 ± 0.49	
Progesterone receptor positive	0.93 ± 0.56	0.020
Progesterone receptor negative	0.61 ± 0.34	

## Data Availability

Not applicable.

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
