# Peer review of "ABCB1* Amplicon Contains Cyclic AMP Response Element-Driven *TRIP6* Gene in Taxane-Resistant MCF-7 Breast Cancer Sublines"

_genes, 2023, doi:10.3390/genes14020296_

Round 1

Reviewer 1 Report

I have only one comment, I suggest that the statement of lines 67 - 69 should go in the discussion

Author Response

Dear Reviewer,

We thank you for your recommendation.

We have moved the statement from lines 67 - 69 to 513 - 515 (Discussion part) as you recommended and changed. the citation order (citation no. 21 changed to 62).

Reviewer 2 Report

In this manuscript, the authors evaluated the factors involved in regulating the high TRIP6 expression in the MCF-7 cell line and two MCF-7 derivative taxane-resistant cell lines. In addition, they assessed clinical breast cancer tissues for TRIP6 mRNA expression and reported an association between the presence of high TRIP6 mRNA levels and the premenopausal status of the patients. The manuscript is very well written and easy to follow. The authors thoroughly examined and evaluated the expression of TRIP6 in the tested cell lines at the RNA and protein levels, assessed the promotor methylation, and evaluated the association between TRIP6 and ABCB gene expression. 

Author Response

Dear Reviewer,

We thank you for your kind words about performed revisions.

Reviewer 3 Report

The article would significantly improve the existing body of evidence in the field of study. However, I would propose the following minor revisions:

Lines 105-108:
The phrasing could be more precise for tissue sampling. Please clarify the exact time frame of tissue sampling around the therapies administered. First, I would describe the origin of the tissue sample – biopsy/surgery? My understanding is that in the NACT group, the tissue that has been examined was the breast tissue extracted during radical mastectomy/conservative surgery (wouldn’t consider relevant the specific surgical procedure, just if it originates from surgical intervention or biopsy). However, the ACT group is a mystery to me. From what I understand, those patients underwent radical surgery and adjuvant chemo/hormonal therapy. So, when was the tissue sampled, and also, what tissue was sampled if there was no more breast cancer tissue to examine post-mastectomy? If the tissue sampling was done before adjuvant therapy, then what is the relevance of the adjuvant therapy? Why haven’t more metastatic patients been included in your study? Did any patients undergo radiation therapy? I’m suggesting that the authors develop more on why this group of patients was selected, how the tissue was sampled and in which time frame regarding systemic therapy.

Furthermore, I would suggest highlighting that the Taxane-FAC regimen was the standard of care in that period (2003 – 2009), as the current guidelines do not support the addition of 5-fluorouracil to the anthracycline (Doxorubicin/Epirubicin) – cyclophosphamide regimen (not mandatory, but valuable for context).

Table 4:
I would classify the tumors as Ki67 high or low (with the appropriate cutoff value) and not Ki67 positive or negative and mention it in the “Collection and processing of breast cancer tissue samples” section.

Also, I would ask about the inclusion of that one patient with Stage IV Breast Cancer included in the study, as patients in this setting are not eligible for either neoadjuvant or adjuvant chemo/hormonal therapy.

One of my curiosities would be why haven’t more patients with Stage IV Breast Cancer been evaluated. Perhaps the metastatic setting, with heavily pre-treated patients, with multiple lines of therapy, might provide more significance to your further research. It is unclear why you have selected this group of neoadjuvant/adjuvant patients, but I am convinced the authors can provide a satisfying argument.

Introduction and Conclusion:
More clinical context is needed to highlight the importance of your study.  Could your findings impact the clinical management of Breast Cancer patients? Perhaps, cost-effectiveness aside, your findings could suggest a possible biomarker discovery. If so, which patients might benefit from this?

Nonetheless, I applaud the authors for this very well-written, well-documented article, which clearly and unequivocally demonstrates their burden of proof!

Author Response

Dear Reviewer,

We kindly thank you for your supportive, mainly positive, feedback on the manuscript. We have revised it to include all the points you suggested and highlighted this study's importance for readers.

Lines 105-108:

The phrasing could be more precise for tissue sampling. Please clarify the exact time frame of tissue sampling around the therapies administered. First, I would describe the origin of the tissue sample – biopsy/surgery? My understanding is that in the NACT group, the tissue that has been examined was the breast tissue extracted during radical mastectomy/conservative surgery (wouldn’t consider relevant the specific surgical procedure, just if it originates from surgical intervention or biopsy). However, the ACT group is a mystery to me. From what I understand, those patients underwent radical surgery and adjuvant chemo/hormonal therapy. So, when was the tissue sampled, and also, what tissue was sampled if there was no more breast cancer tissue to examine post-mastectomy? If the tissue sampling was done before adjuvant therapy, then what is the relevance of the adjuvant therapy? Why haven’t more metastatic patients been included in your study? Did any patients undergo radiation therapy? I’m suggesting that the authors develop more on why this group of patients was selected, how the tissue was sampled and in which time frame regarding systemic therapy.

Furthermore, I would suggest highlighting that the Taxane-FAC regimen was the standard of care in that period (2003 – 2009), as the current guidelines do not support the addition of 5-fluorouracil to the anthracycline (Doxorubicin/Epirubicin) – cyclophosphamide regimen (not mandatory, but valuable for context).

Response: We are indebted to the reviewer for this comment that allows us to explain better the collection procedure, classification, and characterization of breast carcinoma tissue samples in this study. We edited a section describing the collection and processing of breast cancer tissue samples. We clarified the exact time schedule of the tissue sampling and the therapy type and regimens. All changes are tracked by Track Changes in the manuscript in lines 119 - 130.

We also highlighted that Taxane-FAC regimens were used as the standard of care in the period of samples collection and the current guidelines do not support the addition of 5-fluorouracil to the anthracycline (Doxorubicin/Epirubicin) – cyclophosphamide regimen (lines 124 - 126). We have modified the last paragraph in the Discussion section to justify our intention to perform analyses in breast cancer patients undergoing therapy with taxanes (lines 274 - 578).

Table 4:

I would classify the tumors as Ki67 high or low (with the appropriate cutoff value) and not Ki67 positive or negative and mention it in the “Collection and processing of breast cancer tissue samples” section.

Response: Thank you very much for this comment and recommendation. We added information describing the appropriate cutoff value of Ki67, including reference (lines 137 - 138). As the reviewer suggested, we changed the classification of tumors on Ki67 high and low in Table 4 in the manuscript and added a footnoter (line 483).

Also, I would ask about the inclusion of that one patient with Stage IV Breast Cancer included in the study, as patients in this setting are not eligible for either neoadjuvant or adjuvant chemo/hormonal therapy.

One of my curiosities would be why haven’t more patients with Stage IV Breast Cancer been evaluated. Perhaps the metastatic setting, with heavily pre-treated patients, with multiple lines of therapy, might provide more significance to your further research. It is unclear why you have selected this group of neoadjuvant/adjuvant patients, but I am convinced the authors can provide a satisfying argument.

Response: Thank you very much for this comment. Of our available patients, just one was in an advanced stage IV. Reviewer 3 is correct that stage IV patients are a distinct group of breast carcinomas both from the view of prognosis and molecular characteristics. Perhaps interesting for this type of study, but hard to get as fresh frozen specimens in the sampling period. Today, definitely worth trying, but not in the time frame of the present study. Up to that, biopsies may be more often available but are usually scarce in the material content. We primarily focused our study on patients without metastatic spread. We included one stage IV based on treatment with neoadjuvant regimen and subsequent palliative therapy regimen, which contained the same cytotoxic drugs as the rest of the patients. We apologize for incorrect description of this patient in the set, which we corrected in the revised version. We excluded this patient from survival analyses not to mix patients with different prognostic scenarios per se. We would be willing to exclude this patient from the study in case Reviewer 3 considers it necessary. As the reviewer correctly points out, we would like to focus on the collection of specifically stages IV, heavily pre-treated patients and TNBC patients for further research and provide a study separately on their molecular profiles and the role of TRIP6 specifically in those prognostically worse groups of breast carcinoma patients.

We highlighted that patient is included in the Neoadjuvant group (footnote no. 2 in Table 4)

Introduction and Conclusion:

More clinical context is needed to highlight the importance of your study. Could your findings impact the clinical management of Breast Cancer patients? Perhaps, cost-effectiveness aside, your findings could suggest a possible biomarker discovery. If so, which patients might benefit from this?

Response: We are again grateful to Reviewer 3 for sharing his/her broad expertise. For robust conclusions about TRIP6's role in breast cancer, we employed the TCGA, and METABRIC data (mRNA) sets to corroborate our findings. These analyses support the view that TRIP6 is currently not enough proven biomarker in breast cancer to be recommended for clinical applications. However, one previously published study (Zhao et al., 2020; DOI:10.1186/s12935-020-1136-z) provided evidence that TRIP6 might be a novel predictive biomarker and our present findings show that TRIP6 is associated with the progesterone receptor expression status. Thus, follow-up studies need further proof by molecular testing, e.g., with the help of specific inhibitors. Therefore, we have added two paragraphs concerning breast cancer to the Introduction section (lines 75 - 88) and modified the conclusions (lines 598, 602 - 604 and 608 - 610) as well as the citation list (new citations no. 25 to 30).